SCIENCE FORUM

# Improving preclinical studies through replications

**Abstract** The purpose of preclinical research is to inform the development of novel diagnostics or therapeutics, and the results of experiments on animal models of disease often inform the decision to conduct studies in humans. However, a substantial number of clinical trials fail, even when preclinical studies have apparently demonstrated the efficacy of a given intervention. A number of large-scale replication studies are currently trying to identify the factors that influence the robustness of preclinical research. Here, we discuss replications in the context of preclinical research trajectories, and argue that increasing validity should be a priority when selecting experiments to replicate and when performing the replication. We conclude that systematically improving three domains of validity – internal, external and translational – will result in a more efficient allocation of resources, will be more ethical, and will ultimately increase the chances of successful translation.

**NATASCHA INGRID DRUDE[†], LORENA MARTINEZ GAMBOA[†], MEGGIE DANZIGER, ULRICH DIRNAGL AND ULF TOELCH***

## Introduction

Translation from preclinical research to patients is challenging for many reasons (*Denayer et al., 2014*; *Pound and Ritskes-Hoitinga, 2018*). Biological complexity and the disparity between animal models of disease and humans accounts for some failures in translation, but not all (*Kimmelman and London, 2011*; *Mullane and Williams, 2019*; *van der Worp et al., 2010*). Other reasons include the fact that the evidence generated in preclinical efficacy studies is often weak due to the low numbers of experimental units or entities receiving a treatment (*Bonapersona et al., 2019*; *Carneiro et al., 2018*; *Howells et al., 2014*). Moreover, analyses are often not reported in full, leading to the selective reporting of outcomes and the exploitation of researcher degrees of freedom (*Motulsky, 2014*). False positives abound in such studies, and reported effects are often inflated (*Dirnagl, 2020*; *Kimmelman et al., 2014*; *Turner and Barbee, 2019*). Moreover, a strong bias against the publication of non-significant results augments this problem (*Sena et al., 2010*) and makes meta-analytic assessment of preclinical evidence difficult (*Sena et al., 2014*).

An obvious way to address some of these problem would be for other research groups to reproduce and then replicate preclinical studies before starting experiments on humans. By reproduce we mean to be in principle able to repeat the original study through in depth understanding of methods, protocols, and analytical pipelines used by the original research group. By replicate we mean to actually perform a study to see if the findings of the original study still hold. This potentially involves adapting some methods, protocols, and analytical pipelines (*Nosek and Errington, 2020a*; *Patil et al., 2016*): for example, when different animal strains are used or when environmental factors are changed (*Voelkl et al., 2020*). Reproducibility is thus a prerequisite for engaging in replications that will increase our confidence in a finding through its wider validity.

Here, we consider the role of replications in the context of the preclinical research trajectory for a potential treatment: such a trajectory is a series of experiments designed to generate evidence that will inform any decision about testing the treatment in humans. The experiments in a preclinical research trajectory typically include exploratory studies, toxicity studies, positive and

*For correspondence: ulf.toelch@bihealth.de

[†]These authors contributed equally to this work

**Competing interests:** The authors declare that no competing interests exist.

negative controls, pharmacodynamics and kinetics, and are intended to generate evidence to support an inferential claim and refute possible alternatives. They can be performed on animal models (including invertebrates, zebrafish, non-human primates and, quite often, rodents) or with replacement methods (such as cell cultures and organoids).

Within this framework, replications strengthen two key characteristics of preclinical experimental evidence: validity and reliability. Validity refers to the degree to which an inference is true, and reliability refers to the quality and accuracy of the data supporting an inferential claim. In this article we describe strategies for preclinical research trajectories in which replications balance reliability and validity to foster preclinical and translational research in a way that is ethical and efficient. For example, consider a researcher who hypothesizes that a disease is caused by a metabolic product. A potential drug candidate will inhibit an enzyme that is involved in the relevant metabolic process. In an exploratory study, applying the drug in a knockout mouse model of the disease reduced the metabolic product and the health condition of the animals improved. A within-lab replication confirms these initial findings. The findings are reliable as initial data and replication support the inferential claim that the drug improves health conditions. However, this does not necessarily mean that the inference is valid, particularly when extrapolated to humans: as we will outline below, the validity of such an inference can be threatened on several levels during the preclinical research trajectory.

## What to replicate and how to replicate

A number of large-scale replication projects have been conducted in psychology and social sciences (*Camerer et al., 2016*; *Camerer et al., 2018*; *Open Science Collaboration, 2015*). Those studies were performed with healthy subjects with little to no harm anticipated through participation. Consequently, increasing the number of tested subjects was not ethically problematic or overly expensive. The same is not true for projects that involve animals, so there have been relatively few large-scale replication projects in biomedical research. Moreover, the projects that have been started – such as the Reproducibility Project: Cancer Biology, the Brazilian Reproducibility Initiative, and the Confirmatory Preclinical Studies project – all take different approaches to identifying the studies to be replicated and to performing the replications (see *Table 1* and *Box 1*).

Based on the results from the study being replicated and the stage in the preclinical research trajectory, the question is: what additional evidence is needed to ultimately decide to start trials in human subjects? Throughout a sequence of preclinical experiments, validity and reliability have to be adapted at each stage, and criteria are set so that we know whether to continue, to revise, or even completely break off the experiments (*Figure 1*). How, for example, could

**Table 1.** Overview of three large-scale replication projects in biomedical research: Reproducibility Project: Cancer Biology (RPCB); Brazilian Reproducibility Initiative (BRI); Confirmatory Preclinical Studies (CPS).

|  | RPCB | BRI | CPS |
|---|---|---|---|
| Selection of samples to be replicated | Main findings from 50 high impact citations/publications in cancer research | Replication of 60–100 experiments from research articles of Brazilian studies in different clinical areas | Two-step review process of proposals results in twelve projects |
| Selection of experiment | Main finding from published studies | Experiments using five pre-defined methods | Own experiments |
| Replicate own results | No | No | Yes |
| Exact Protocols | Yes (consulting original authors) | No | Yes |
| Blind to initial results | No | Yes | No |
| Pre-registration | Pre-registered study and individual Replication Protocols | Yes | Yes |
| Multi-site replication | No | Yes | Yes |

## Box 1. Three approaches to large-scale replication projects in biomedical research.

The Reproducibility Project: Cancer Biology (RPCB) started with the aim of reproducing selected findings from 50 high-impact articles published between 2010 and 2012 in the field of cancer biology (*Errington et al., 2014*; *Morrison, 2014*). The plan was to publish a peer-reviewed Registered Report that outlined the protocols for each attempted reproduction – based on information contained in the original paper and, if necessary, additional information obtained from the original authors – before any experiments were performed (*Nosek and Errington, 2020b*). The experiments were to be conducted by commercial contract research organizations and academic core facilities from the Science Exchange network, and the results were to be published in a separate peer-reviewed Replication Study. The researchers performing the experiments were not blinded with regard to the original results. In the end, due to various problems, only 29 Registered Reports and 18 Replication Studies were published, and the overall conclusions of the project are currently being written up. The aim of the Brazilian Reproducibility Initiative is to assess the reproducibility of biomedical science published by researchers based in Brazil (*Amaral et al., 2019*; *Neves et al., 2020*). The studies selected had to use one of five experimental techniques, including behavioural and wet lab methods, on certain widely-used model organisms. The BRI researchers assume that protocols will never be reproduced exactly so they employ a 'naturalistic approach' in which the teams repeating the experiments can supplement the published protocols based on their best judgement and experience. Moreover, three teams will attempt to repeat each study selected, and will pre-register their protocols before starting experiments. Furthermore, the researchers performing the experiments will be blinded to the identity of the original authors and the results of the paper. Recently, the Federal Ministry of Education and Research in Germany invited research groups to apply for funding to attempt to confirm promising results from their own preclinical studies (*BMBF-DLR, 2018*). After being screened and reviewed by a panel of international experts, 12 groups have received funding under the CPS (Confirmatory Preclinical Studies) project: one condition of the project is that the groups funded have to collaborate with other groups (of their choosing) in a multi-centre approach with a view to harmonising protocols across sites. Again the groups will have to pre-register their protocols: however, as the researchers are repeating their own experiments, they will not be blinded to the original results.

the previous experiments be improved? How many animals should be tested? Are additional controls needed? Should additional labs be involved? At all stages the aim should be to increase the validity of replications and test the reliability of the initial finding (*Dirnagl, 2020*; *Kimmelman et al., 2014*; *Piper et al., 2019*).

Researchers usually start off with a study in exploratory mode. As not all details and confounders in such a study can be known upfront, the reliability and validity are potentially not fully optimized. However, even at these early stages researchers should implement strategies to mitigate risks of bias (*Figure 1*). After an initial

phase, the question is: should we continue to test a particular claim? The criteria used to answer this question should be lenient: standard p-value criteria are potentially too strict and identifying the range of possible effect sizes is a viable goal in early phases. Particularly in fields with a low prevalence of true hypotheses, this will prevent discarding promising treatments too early (*Albers and Lakens, 2018*; *Lakens, 2014*). More stringent criteria should be applied in later experiments. Standard p-values will then identify true effects in high powered replications, which will reduce Type I errors and increase the predictive value of a set of experiments.

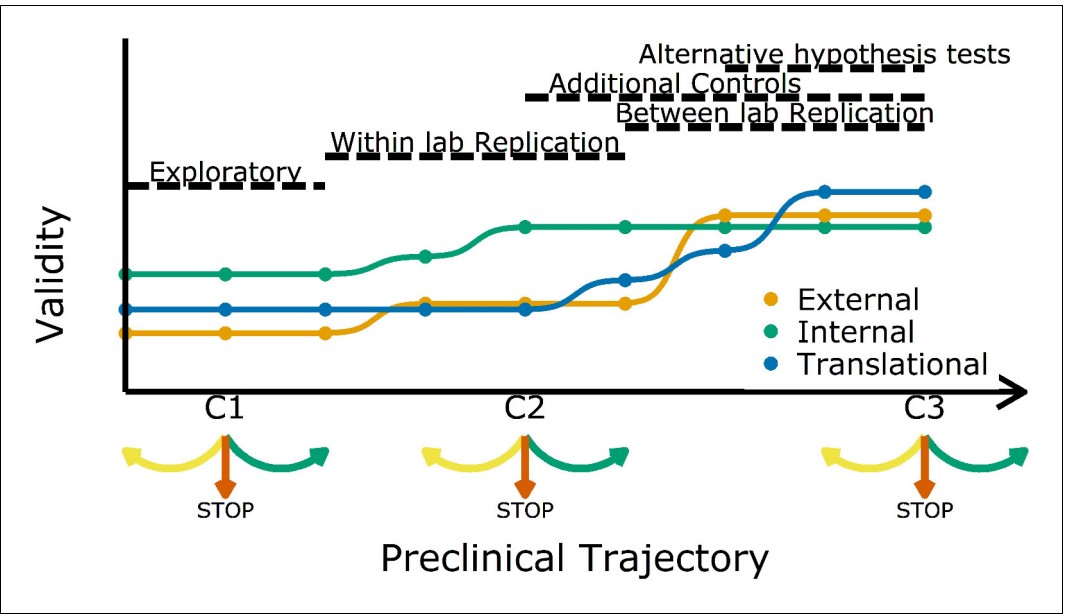

**Figure 1.** Increasing different forms of validity (internal, external and translational) during a preclinical research trajectory. This schematic shows that internal validity (green line) is higher than external validity (orange) and translational validity (blue) at the start of a preclinical research trajectory (left), and that evidence from different types of experiments can increase different types of validity. For example, evidence from exploratory studies can increase internal and external validity, and evidence from between-lab replications can increase translational validity. C1, C2 and C3 are decision points where researchers can decide to refine the current experiment (yellow arrow), stop the trajectory (red arrow), or proceed to the next experiment (green arrow).

As we move along the research trajectory it also becomes helpful to think in terms of three types of validity: internal validity, external validity, and translation validity (*Figure 1*). To illustrate this, we return to the example of a disease caused by a metabolic product. An experiment in which the inhibition of a metabolic pathway leads to an improvement in animal health is internally valid if the measured effect (improved health) is caused by the experimental manipulation (administration of the inhibitor). Such an experiment is also externally valid if the effect is observed in other animal models and/or can be replicated in other labs. And if the effect is also observed in humans it is translationally valid. As we shall discuss, different strategies are needed to improve these three types of validity.

### Internal validity

Poor study design and the lack of control for biases are major contributors towards low internal validity (*Bailoo et al., 2014*; *Pound and Ritskes-Hoitinga, 2018*; *Würbel, 2017*). To increase internal validity, controlling for selection and detection bias is essential. For this, methods of randomization and blinding need to be clearly specified, and inclusion and exclusion criteria

need to be defined, before data are collected (*Würbel, 2017*). Replications can be leveraged to improve this further. Imagine, for example, that an exploratory study has found that a physiological factor (such as animal weight) influences the primary outcome variable: in the next round of experiments, animals could be randomized and stratified by weight. Particularly for replications, all these choices (along with detailed methods and analysis plans) should be communicated before conducting the experiment, ideally via pre-registration at a platform such as http://www.animalstudyregistry.org or http://www.preclinicaltrials.eu.

Methods and analyses also need to be reported transparently and completely (*Vollert et al., 2020*; *Percie du Sert et al., 2020*). However, completely reported experiments can still have low internal validity because following a reporting guideline will not safeguard against suboptimal experimental design: that is, experiments need high internal validity and complete reporting to prevent research waste and fruitless animal testing (*Macleod et al., 2014*).

The three replication projects described in *Box 1* all aim at high internal validity through

exact specifications of replication protocols. Within the RPCB, for example, protocols were pre-registered, peer reviewed and published before replication experiments were performed. However, in some cases where results have been difficult to interpret, exact pre-specification of replication experiments has limited the possibility of performing the experiments in another, potentially improved way (*eLife, 2017*). The BRI describes a more naturalistic approach where each participating lab will fill the gaps in papers as best as they can without consulting primary authors. Nevertheless, protocols will still be pre-registered and undergo a round of internal peer review among collaborating labs. In summary, high levels of methodological rigour are necessary to ensure high internal validity and to make replications meaningful.

### External validity

For the assessment of external validity, research findings from one setting need to generalise to other settings (*Pound and Ritskes-Hoitinga, 2018*). One way to increase external validity is to conduct replications at multiple sites, emulating an approach already applied in clinical trials (*Dechartres et al., 2011*; *Friedman et al., 2015*). Multi-centre studies (or between-lab replications at a single centre) can index known and unknown differences that define boundary conditions for investigated effects (*Glasgow et al., 2006*; *Pound and Ritskes-Hoitinga, 2018*). Regarding generalizability, external validity can often be improved by including aged or comorbid animals, and by performing multimodal studies with animals of different sex and/or varying strains. This systematically introduced heterogeneity strengthens external validity and explores the extent to which standardization is introducing unwanted idiosyncrasies that may limit external validity and prevent successful replication (*Richter et al., 2010*; *Voelkl et al., 2018*). Additionally negative and positive control groups that are added to replications can further foster external validity (*Kafkafi et al., 2018*).

Again, the three replication projects described in *Box 1* take different approaches. The BRI and CPS projects take multi-centre approaches, whereas the RPCB does not, which may result in lower external validity. However, multi-centre studies have their own limitations and shortcomings as they come with an organizational overhead that includes decisions on which parts of experiments should be standardised and safeguarding adherence to the agreed protocols throughout a study

(*Maysami et al., 2016*). This can be challenging, given different infrastructure and/or resources across laboratories with different budget or resource constraints.

Difference in equipment can be one reason for variation in performing a certain intervention (e.g. surgery). Moreover, if the centres are in different countries, ethics boards and local regulations will most likely differ, complicating and potentially delaying ethics approval (*Hunniford et al., 2019*; *Llovera et al., 2015*; *Maysami et al., 2016*). For example, regulations for analgesic regimes differ between jurisdictions, so that it is difficult to follow the same protocol across sites. Nevertheless, multi-centre studies are characterised by high quality standards with cross validation of results, larger sample size, lower risk of bias as compared to single-centre studies, and higher completeness of reporting (*Hunniford et al., 2019*). This strongly suggests that a multi-centre approach will be an important component in enabling decisions about clinical trial initiation (*Prohaska and Etkin, 2010*).

### Translational validity

Translational validity is used here as an umbrella term for factors that putatively contribute to the translation from animal models to humans. In particular, it pertains to how well measurements and animal models represent a certain disease and its underlying pathomechanisms in humans, as it is common for only a limited number of disease characteristics to present in animal models. In models of Alzheimer's disease, for example, the focus on familial early onset genes in mouse models has potentially led to translational failures as the majority of diagnoses of Alzheimer's disease in humans are classified as sporadic late onset form (*Mullane and Williams, 2019*; *Sasaguri et al., 2017*). Translational validity thus reflects whether measured parameters in animal models are diagnostic for human conditions and consequently, to what extent the observed outcomes will predict outcomes in humans (*Denayer et al., 2014*; *Mullane and Williams, 2019*).

Ideally, auxiliary measures are collected alongside the primary outcome variable of the initial study. While such secondary outcomes might not be recorded in early experiments, replications are ideally suited to including these additional measures. Clinical biomarkers that are diagnostic for a disease in humans can provide information on the translational potential if collected also in replications in animal models

(*Metselaar and Lammers, 2020*; *Volk et al., 2015*). In this context, if an imaging method like MRI is used in the diagnosis of humans, the same method applied in animals can reveal whether physiological parameters and disease location are comparable. This can be combined with experiments that gather converging and discriminant evidence to identify mechanistic underpinnings of an intervention to increase translational validity and identify limitations and boundary conditions. For example, studies of pharmacodynamics and kinetics in preclinical models support in-depth understanding of physiological processes and allow comparison with human pharmacological processes (*Salvadori et al., 2019*; *Tuntland et al., 2014*).

Replication studies can also be performed in a more complex animal model. In cancer research, for example, the initial study might be performed in an animal model with a subcutaneous tumour, while the replication could be conducted in a more advanced tumour model, in which the development of an organ-specific tumour microenvironment more closely mimics the clinical reality (*Guerin et al., 2020*). The decision on which additional information will be helpful should be based on an exchange between preclinical researchers and clinicians.

Therefore, translational validity needs to be considered at each stage during preclinical research.

## Ethical conduct of replications

To optimize evidence from experiments on animals it is necessary to balance the different types of validity and reliability of experiments and replications. Early on, internal validity needs to be established with high priority. As knowledge about the animal model and disease mechanisms increases, external validity needs to be strengthened through within-lab replications and, for core results, to multiple centres. Systematic heterogeneity (additional strains, similar animal models, different sexes) will further strengthen external validity, and such heterogeneity should be introduced at the early stages of the work if this is feasible.

Replications at such later stages should also include secondary outcomes that directly link to clinically relevant parameters. However, even in high-validity experiments, reliability can be low when the number of experimental units is not sufficient to detect existing effects. For replications at this stage, reliability should be increased by increasing sample sizes or refining measurement procedures. As true preclinical effect sizes are frequently small and associated with considerable variance between experimental units, increased numbers of experimental units are needed to obtain reliable results (*Bonapersona et al., 2020*; *Carneiro et al., 2018*). According to the 3R principles (*Russell and Burch, 1959*), the number of animals tested needs to reflect the current stage in the preclinical trajectory (*Sneddon et al., 2017*; *Strech and Dirnagl, 2019*).

There is, however, no consensus yet on how to balance ethical and statistical power considerations in replications in animal experiments. Standard approaches where power calculations are based on the point estimate of the initial study will often yield too small animal numbers (*Albers and Lakens, 2018*; *Piper et al., 2019*). This potentially inflates false negatives, running the risk of missing important effects and wrongfully failing a replication. Alternatives like safeguard power analysis (*Perugini et al., 2014*), sceptical p-value (*Held, 2020*), or adjusting for uncertainty (*Anderson and Maxwell, 2017*) have been proposed mainly for psychological experiments with human subjects. These approaches will often yield high enough sample sizes to ensure sufficient power for replications. Due to ethical and resource constraints, preclinical replications are seldom able to test such high numbers, which may be one reason why such approaches have not yet been implemented widely in preclinical research design. Here, we see clear room for improvement and research opportunities.

Regarding the number of experimental units, the RPCB aimed at achieving at least 80% statistical power, based on the effect size measured in the original study. However, the 'winner's curse' means that published effect sizes (p<0.05) tend to be larger due to random sample variability. So basing the design of a replication on the effect size reported in the original study could result in the replication being underpowered (*Colquhoun, 2014*).

The BRI team calculated sample sizes to achieve a statistical power of 95% to detect the original effect in each of the three replications (which will be conducted in different labs). Through this, the BRI team tried to compensate for a possible inflation of the original results due to publication bias and winner's curse. Furthermore, they planned to include additional positive and/or negative controls to ensure interpretation of the outcomes (*Neves and Amaral, 2020*). The different approaches taken

by RPCB and BRI also confirm that there is, as yet, no consensus on how to calculate animal numbers towards an ethical conduct of replications.

## Summary and recommendations

The goal of a preclinical research trajectory is to enable the decision to engage in clinical studies. In a simplified scheme, decision options include to discontinue experiments because of futility (hypothesis apparently not true or effects not biologically significant), to gather more evidence to resolve ambiguity (with increased validity and reliability), or to engage in a clinical study (when enough evidence is collected). Most studies in preclinical research, however, are targeted at initial findings that are often exploratory (*Howells et al., 2014*). Systematic replication efforts that are decision-enabling are rare in academic preclinical research and have only recently begun to be conducted (*Kimmelman et al., 2014*). For the decision to finally engage in a clinical trial, a systematic review of all experiments is needed. In such a review, evidence should be judged on the validity and reliability criteria discussed here. Ideally, this will form the basis for informative investigator brochures that are currently lacking such decision enabling information (*Wieschowski et al., 2018*).

Our proposed framework is of course not applicable in all cases. If prior knowledge about a mechanism is already available and models are established, internal validity may already be high from the onset. Moreover, our simplified proposal may not generalise across all fields, or to academic and industry settings alike. Nonetheless, replications (and ideally every preclinical experiment) need to be framed in terms of validity and reliability. This is even more pressing as replications constitute an important foundation for successful translation. To enable the establishment of preclinical research trajectories, we see a need for action for funders and researchers.

The surprising lack of systematic replication in preclinical research also stems from lack of funding opportunities. Contrary to clinical trials, across-lab multi-centre replications are rare in preclinical research. Funders should thus design specific calls aimed at replications in the broad sense described here. Funding schemes should take the structure of preclinical research trajectories into account and may specifically be tailored towards the different experimental stages.

For replications, researchers need to specify how validity is improved by the replication compared to an initial study. Detailing how internal, external and translational validity increase in replications will emphasise the new evidence that is generated beyond the initial study and provide an ethical justification for the replication. Sample-size calculations should consider how reliability and validity are balanced against each other and define clear criteria for decisions to advance to the next stage in the trajectory. This will require a deviation from standard one-size-fits-all sample-size calculations. Researchers need guidance on how to adjust sample-size calculations and decision criteria, starting from an initial exploratory study that will serve as a proof of concept and gradually moving towards decision-enabling studies that will define whether a clinical trial is warranted.

The scientific endeavour is not limited to a single study and simple null-hypothesis testing. Even though the prevailing statistical test framework may suggest this, researchers are operating in a larger framework where evidence is accumulated over several levels with several competing alternative hypotheses (*Platt, 1964*). Replications are an important building block, where research priorities are transparently updated according to current knowledge. This includes proper reporting at all stages and registration of research at critical stages to avoid biases (*Strech and Dirnagl, 2019*). Currently, the literature on preclinical research and associated decisions is scarce (see, for example, *van der Staay et al., 2010*). Research on successful – and also on less successful – research lines will inform about best practices and yield important insights how biases potentially distort evidence collection (*Kiwanuka et al., 2018*).

In conclusion, systematically improving scientific validity in replications will improve trustworthiness and usefulness of preclinical studies and thus allow for a responsible conduct of animal experiments.

**Natascha Ingrid Drude** is in the Department of Experimental Neurology, Charité–Universitätsmedizin Berlin and the BIH QUEST Center for Transforming Biomedical Research, Berlin Institute of Health, Berlin, Germany
https://orcid.org/0000-0002-7153-2894

**Lorena Martinez Gamboa** is in the Department of Experimental Neurology, Charité–Universitätsmedizin Berlin and the BIH QUEST Center for Transforming Biomedical Research, Berlin Institute of Health, Berlin, Germany

**Meggie Danziger** is in the Department of Experimental Neurology, Charité–Universitätsmedizin Berlin and the BIH QUEST Center for Transforming Biomedical Research, Berlin Institute of Health, Berlin, Germany

https://orcid.org/0000-0001-9224-1722

**Ulrich Dirnagl** is in the Department of Experimental Neurology, Charité–Universitätsmedizin Berlin and the BIH QUEST Center for Transforming Biomedical Research, Berlin Institute of Health, Berlin, Germany

https://orcid.org/0000-0003-0755-6119

**Ulf Toelch** is in the BIH QUEST Center for Transforming Biomedical Research, Berlin Institute of Health, Berlin, Germany

ulf.toelch@bihealth.de

https://orcid.org/0000-0002-8731-3530

*Author contributions:* Natascha Ingrid Drude, Conceptualization, Writing - original draft, Writing - review and editing; Lorena Martinez Gamboa, Conceptualization, Writing - original draft, Project administration, Writing - review and editing; Meggie Danziger, Conceptualization, Writing - review and editing; Ulrich Dirnagl, Conceptualization, Funding acquisition, Writing - original draft, Writing - review and editing; Ulf Toelch, Conceptualization, Visualization, Writing - original draft, Writing - review and editing

*Competing interests:* The authors declare that no competing interests exist.

## Funding

| Funder | Grant reference number | Author |
| --- | --- | --- |
| Bundesminister-ium für Bildung und Forschung | 01KC1901A | Natascha Ingrid Drude Lorena Martinez Gamboa |

The funders had no role in study design, data collection and interpretation, or the decision to submit the work for publication.

**Decision letter and Author response**
Decision letter https://doi.org/10.7554/eLife.62101.sa1
Author response https://doi.org/10.7554/eLife.62101.sa2

## Additional files

### Data availability

No data was generated.

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
