## [Decision Letter]

Thank you for submitting your article "Improving the predictiveness and ethics of preclinical studies through replications" to *eLife* for consideration as a Feature Article. Your article has been reviewed by three peer reviewers, and the evaluation has been overseen by the *eLife* Features Editor (Peter Rodgers). The following individuals involved in review of your submission have agreed to reveal their identity: Catherine Winchester (Reviewer #1); Hanno Wuerbel (Reviewer #2).

The reviewers and editors have discussed the reviews and we have drafted this decision letter to help you prepare a revised submission.

Summary

This paper presents an informal framework to guide preclinical research programmes, with the aim to systematically and gradually improve the reliability and validity of the research findings. According to the authors, such a procedure would not only increase chances for translational success but also lead to a more efficient allocation of resources and ethical use of animals. While the overall message is clear – most experiments are part of larger research programmes, and experimental designs need to be adjusted to the specific questions asked at any stage of a research programme – the authors need to be more precise in their definition/use of terms like replication, and to provide more details on how their proposals would work in practice.

Essential revisions

1) Definition of replication

The definition of "replication" used in the article seems almost redundant with the term "experiment". For example, the authors state: "Here, we define replications as experiments that aim to generate evidence that supports a previously made inferential claim (Nosek and Errington, 2020a)".

First, I am not sure this is equivalent to definition given in Nosek and Errington, 2020a: "Replication is a study for which any outcome would be considered diagnostic evidence about a claim from prior research". This should be clarified.

However, even if so, I am still not convinced it would be a useful definition of "replication". Nosek and Errington conclude: "Theories make predictions; replications test those predictions. Outcomes from replications are fodder for refining, altering, or extending theory to generate new predictions. Replication is a central part of the iterative maturing cycle of description, prediction, and explanation."

This is in fact a concise description of the "scientific method". Thus, Wikipedia (sorry about that….) describes the "scientific method" as: "It [the scientific method] involves formulating hypotheses, via induction, based on such observations; experimental and measurement-based testing of deductions drawn from the hypotheses; and refinement (or elimination) of the hypotheses based on the experimental findings."

I don't think it makes sense to replace well established terms such as "scientific method" or "experiment" by "replication", which for most scientists has a much more specific meaning. The authors should clarify this point and clearly state how they distinguish their term "replication" from the term "experiment" (every experiment is a test of a claim from prior research and therefore fits Nosek and Errington's definition of replication) and how they distinguish their framework from the "scientific method" in general.

This becomes even more explicit in the following paragraph, where they state: "Besides within and between laboratory replications, experiments in such series include initial exploratory studies, toxicity studies, positive and negative controls, pharmacodynamics and kinetics that all aim to generate evidence to support an inferential claim and refute possible alternatives". Thus, according to the authors, all of these experiments (toxicity studies, pharmacodynamics, etc.) are nothing but replications – this is stretching the term replication extremely far, and I am not convinced that readers will follow them. And it would not be necessary; they could present the exact same framework by using conventional terms.

Related to this, please explain the difference/relationship between replication and reproducibility.

2) The section "What to replicate?"

The description of the three projects in the section is too long, and needs to be better integrated into the article.

Note from the Editor: since this point is largely an editorial matter, dealing with it can be deferred until later.

3) The section "How to replicate?"

The stepwise procedure presented in the text and in Figure 1 is not well developed. The trade-offs between the different dimensions of validity are not presented in any detail, and the ethical and scientific implications of different decisions at different stages in a research programme are not discussed. In particular:

– "At the start, reliability and validity are potentially low due to biased assessment of outcomes, unblinded conduct of experiments, undisclosed researcher degrees of freedom, low sample sizes, etc.". This reads like a recommendation: start your research line with quick and dirty studies and only refine methods once you have discovered something interesting. I am sure that this is not what the authors wanted to say but it is how it may come across. The fact that these problems are prevalent throughout past (and current) preclinical research does not justify them. Even at early, exploratory stages of a research programme, scientific rigour is essential for both scientific and ethical reasons. Therefore, I do not agree with the authors that internal validity may be low initially and should only be improved in the course of a research programme. Studies may be smaller (using smaller sample sizes) initially, but there is no excuse for ignoring any other measure against risks of bias (randomization, blinding, complete reporting of all measured outcome variables, etc.) as these are simply part of good research practice!

– This brings me to another point: I am not convinced that exploratory research should be conducted under highly restricted conditions. In fact, including variation of conditions (e.g. genotype, environment, measurement) right from the beginning may be much more productive a strategy for generating valuable hypotheses (see e.g. Voelkl et al., 2020). Furthermore, designing larger and more complex confirmatory or replication studies (covering a larger inference space) may reduce the need for additional studies to assess convergent and discriminant validity and external validity. Thus, there is a trade-off between more but simpler vs. fewer but more complex studies that is not at all represented in the framework presented here (and in Figure 1).

– "At this stage, criteria to decide whether to conduct additional experiments and replications should be lenient". Here it seems that the authors do make a difference between "experiments" and "replications", but as discussed above, it is unclear on what grounds.

– The definition of "translational validity" needs attention. They state that it covers "construct validity" and "predictive validity", but not "concurrent validity" and "content validity", which are the four common dimensions of test validity. Thus, it is unclear whether they actually mean "test validity" (but consider concurrent validity and content validity less important) or whether they consider "translational validity" to be different from "test validity". Importantly, however, "construct validity" is made up by "convergent validity" and "discriminant validity". This means that assessing "construct validity" requires tests of convergent and discriminant validity that cannot – according to common terminology – be considered as replications as they explicitly test and compare measures of different constructs that should or should not be related.

– "In studies with low internal validity that stand at the beginning of a new research line, e.g. in an exploratory phase, low numbers of animals may be acceptable." Again, I don't think low internal validity is ever acceptable, certainly not if animals are involved. The number of animals always needs to be justified by the specific question to be answered – that in exploratory research often low numbers of animals are used does not necessarily mean that this is acceptable.

– The recommendations in this section are rather vague. For example, the authors write: "For replications, researchers need to specify how validity is improved by the replication compared to the initial study." Yet, after this general demand, no further details or suggestions are given how this can be achieved. Similarly, they write "Researchers need guidance on how to adjust sample size calculations..." but no guidance is offered here nor are any hints given where to find such guidance (references might be useful) on what principles this guidance should be based etc.

– Please comment on how much evidence (how many replication studies/ different approaches exploring the same question) is needed? And who should collate this body of evidence – researchers, clinicians, funders, a committee?

---

## [Author Response]

Essential revisions1) Definition of replicationThe definition of "replication" used in the article seems almost redundant with the term "experiment". For example, the authors state: "Here, we define replications as experiments that aim to generate evidence that supports a previously made inferential claim (Nosek and Errington, 2020a)".First, I am not sure this is equivalent to definition given in Nosek and Errington, 2020a: "Replication is a study for which any outcome would be considered diagnostic evidence about a claim from prior research". This should be clarified.However, even if so, I am still not convinced it would be a useful definition of "replication". Nosek and Errington conclude: "Theories make predictions; replications test those predictions. Outcomes from replications are fodder for refining, altering, or extending theory to generate new predictions. Replication is a central part of the iterative maturing cycle of description, prediction, and explanation."This is in fact a concise description of the "scientific method". Thus, Wikipedia (sorry about that….) describes the "scientific method" as: "It [the scientific method] involves formulating hypotheses, via induction, based on such observations; experimental and measurement-based testing of deductions drawn from the hypotheses; and refinement (or elimination) of the hypotheses based on the experimental findings."I don't think it makes sense to replace well established terms such as "scientific method" or "experiment" by "replication", which for most scientists has a much more specific meaning. The authors should clarify this point and clearly state how they distinguish their term "replication" from the term "experiment" (every experiment is a test of a claim from prior research and therefore fits Nosek and Errington's definition of replication) and how they distinguish their framework from the "scientific method" in general.This becomes even more explicit in the following paragraph, where they state: "Besides within and between laboratory replications, experiments in such series include initial exploratory studies, toxicity studies, positive and negative controls, pharmacodynamics and kinetics that all aim to generate evidence to support an inferential claim and refute possible alternatives". Thus, according to the authors, all of these experiments (toxicity studies, pharmacodynamics, etc.) are nothing but replications – this is stretching the term replication extremely far, and I am not convinced that readers will follow them. And it would not be necessary; they could present the exact same framework by using conventional terms.Related to this, please explain the difference/relationship between replication and reproducibility.

We agree that our definition was not exactly matching the Nosek and Errington definition. We now clarify that replications are not solely about supporting evidence but works in both ways also to refute a claim. Starting from the admittedly very broad Nosek and Errington definition, we now specify that a replication is indeed based on a specific previous experiment. We thus rewrote the passage mentioned above to clarify this important issue. We now distinguish between experiments that test the same claim but with different design and approach and replications that are closely modelled after an initial experiment, but are changed to improve validity and reliability.

“Here, we define replications as experiments that are based on previous studies that aim to distinguish false from true claims (Nosek and Errington, 2020a). For this, previous experiments need to be reproducible with all methods and analytical pipelines unambiguously described. Reproducibility is thus a necessary prerequisite to engage in a contrastable replication (Patil et al., 2016; Plesser, 2018). Replications can deviate from previous experimental protocols by e.g. introducing different animal strains or changing environmental factors. Introducing such systematic heterogeneity between studies potentially strengthens generated evidence about inferential claims (Voelkl et al., 2020). “

With regard to the scientific method, we now emphasise that even tough preclinical research is of course based on a scientific method we now emphasise distinct features (measuring human conditions entirely in model systems) that sets it apart from others disciplines.

“With its goal to closely model human disease conditions, preclinical research differs from other scientific disciplines as there is a biological gap between experimentally studied models and patients as the ultimate beneficent. This affects the role of replications as they should not only confirm previous results but ideally also increase predictive power for the human case to enable successful translation.”

2) The section "What to replicate?"The description of the three projects in the section is too long, and needs to be better integrated into the article.Note from the Editor: since this point is largely an editorial matter, dealing with it can be deferred until later.

We make a suggestion how to shorten this paragraph. We reduced the number of words from 1027 to 807. We look forward to discussing this paragraph further if necessary.

3) The section "How to replicate?"The stepwise procedure presented in the text and in Figure 1 is not well developed. The trade-offs between the different dimensions of validity are not presented in any detail, and the ethical and scientific implications of different decisions at different stages in a research programme are not discussed. In particular:– "At the start, reliability and validity are potentially low due to biased assessment of outcomes, unblinded conduct of experiments, undisclosed researcher degrees of freedom, low sample sizes, etc.". This reads like a recommendation: start your research line with quick and dirty studies and only refine methods once you have discovered something interesting. I am sure that this is not what the authors wanted to say but it is how it may come across. The fact that these problems are prevalent throughout past (and current) preclinical research does not justify them. Even at early, exploratory stages of a research programme, scientific rigour is essential for both scientific and ethical reasons. Therefore, I do not agree with the authors that internal validity may be low initially and should only be improved in the course of a research programme. Studies may be smaller (using smaller sample sizes) initially, but there is no excuse for ignoring any other measure against risks of bias (randomization, blinding, complete reporting of all measured outcome variables, etc.) as these are simply part of good research practice!

We fully agree with the reviewers here and admit that we introduced some ambiguity here that we now clarify. In particular, we stress that internal validity should be already high in early stages. We also adjusted Figure 1 to indicate that internal validity should be already considered at early stages.

“At the start, researchers usually start off with an explorative study. As not all details and confounders in such a study can be known upfront reliability and validity are potentially not fully optimized. However, even at these early stages researchers should implement e.g. strategies to mitigate risks of bias (Figure 1).”

– This brings me to another point: I am not convinced that exploratory research should be conducted under highly restricted conditions. In fact, including variation of conditions (e.g. genotype, environment, measurement) right from the beginning may be much more productive a strategy for generating valuable hypotheses (see e.g. Voelkl et al., 2020). Furthermore, designing larger and more complex confirmatory or replication studies (covering a larger inference space) may reduce the need for additional studies to assess convergent and discriminant validity and external validity. Thus, there is a trade-off between more but simpler vs. fewer but more complex studies that is not at all represented in the framework presented here (and in Figure 1).

We agree with the reviewers that the standardisation fallacy is an important topic, particularly to be considered in preclinical research. As our focus is on the role of replications, we now extend our description on within and between laboratory replications to consider heterogenization. We emphasise this at various locations now. We further point readers to the mentioned paper by Voelkl et al., 2020 as a reference for a more general discussion. At this stage, we feel that introducing also a discussion on how to start a preclinical research process and negotiate simpler vs more complex studies would stray too far from our core message. To accommodate this important thought nonetheless we now state:

“Systematic heterogeneity (additional strains, similar animal models, different sex) will further strengthen external validity. If feasible heterogenization should be introduced at early stages already.”

– "At this stage, criteria to decide whether to conduct additional experiments and replications should be lenient". Here it seems that the authors do make a difference between "experiments" and "replications", but as discussed above, it is unclear on what grounds.

We adjusted the Introduction (see above) to make the distinction between a replication and additional experiments in preclinical research clear.

– The definition of "translational validity" needs attention. They state that it covers "construct validity" and "predictive validity", but not "concurrent validity" and "content validity", which are the four common dimensions of test validity. Thus, it is unclear whether they actually mean "test validity" (but consider concurrent validity and content validity less important) or whether they consider "translational validity" to be different from "test validity". Importantly, however, "construct validity" is made up by "convergent validity" and "discriminant validity". This means that assessing "construct validity" requires tests of convergent and discriminant validity that cannot – according to common terminology – be considered as replications as they explicitly test and compare measures of different constructs that should or should not be related.

We rewrote the paragraph on translational validity to address these points. Importantly, we removed all references to other types of validity than translational validity. The types of validity cited are derived mainly from psychological test theory. They have specific meanings in this context and it is (after lengthy discussion amongst the authors) not clear whether they relate to preclinical research in the same way they relate to psychological test theory. Even in psychology, construct validity is still a hotly debated topic where Meehl/Cronbach accounts clash with Messick for example. As this paper is addressing preclinical researchers who will most likely not be familiar with these terms and their theoretical framing, we removed the terms. We retained the original implications though. That is, translational validity is about the appropriateness of the disease model on a mechanistic level and how well it predicts the human condition. We also introduced as suggested by the reviewers converging and discriminant evidence without however resorting to calling this a type of validity. We clarify also that experiments generating converging and discriminant evidence are separate from replications.

“The term translational validity is used here as an umbrella term for factors that putatively contribute to translational success. It pertains to how well measurements and animal models represent a certain disease and its underlying pathomechanisms in humans. To assess this, complementary experiments evaluate the bounds of a model to discriminate it from other very similar diseases and collect converging evidence from different approaches. Often only parts of a disease are present in animal models. In models of neurodegenerative diseases like Alzheimer’s disease (AD) (Sasaguri et al., 2017), the focus on familial early onset genes in mouse models has potentially led to translational failures as the majority of human AD diagnoses are classified as sporadic late onset form (Mullane and Williams, 2019). Translational validity thus reflects whether measured parameters in animal models are diagnostic for human conditions and consequently, to what extent the observed outcomes will predict outcomes in humans (Denayer et al., 2014; Mullane and Williams, 2019).”

– "In studies with low internal validity that stand at the beginning of a new research line, e.g. in an exploratory phase, low numbers of animals may be acceptable." Again, I don't think low internal validity is ever acceptable, certainly not if animals are involved. The number of animals always needs to be justified by the specific question to be answered – that in exploratory research often low numbers of animals are used does not necessarily mean that this is acceptable.– The recommendations in this section are rather vague. For example, the authors write: "For replications, researchers need to specify how validity is improved by the replication compared to the initial study." Yet, after this general demand, no further details or suggestions are given how this can be achieved. Similarly, they write "Researchers need guidance on how to adjust sample size calculations..." but no guidance is offered here nor are any hints given where to find such guidance (references might be useful) on what principles this guidance should be based etc.

Here, we want to point out that the number of animals is in our view not so much related to validity as to reliability. We now address this point in more detail (the reviewers note themselves above: “Studies may be smaller (using smaller sample sizes) initially.”).

“However, even in high-validity experiments, reliability can be low when the number of experimental units is not sufficient to detect existing effects. For replications at this stage, reliability should be increased by increasing sample sizes or refining measurement procedures. As true preclinical effect sizes are frequently small and associated with considerable variance between experimental units, increased numbers of experimental units are needed to obtain reliable results (Bonapersona et al., 2020; Carneiro et al., 2018). “

We further provide concrete examples how to increase internal validity, external validity and translational validity.

Guidance on sample size calculations in replications is scarce and a theme that we are currently researching ourselves. We nonetheless now point to recent approaches that were developed mainly for psychology and thus may not transfer easily.

“Under consideration of a reduction of animals according to the 3R principles (Russell and Burch, 1959), the number of animals tested needs to reflect the current stage in the preclinical trajectory (Sneddon et al., 2017; Strech and Dirnagl, 2019). For replications, this estimation involves consideration of effect sizes from previous studies. A power calculation based on the point estimate of the initial study will often yield too small animal numbers (Albers and Lakens, 2018; Piper et al., 2019). This potentially inflates false negatives, running the risk of missing important effects and wrongfully failing a replication. Alternatives like safeguard power analysis (Perugini et al., 2014), sceptical p-value (Held, 2020), or adjusting for uncertainty (Anderson and Maxwell, 2017) have been proposed mainly for psychological experiments with human subjects. Sample sizes in animal experiments are, however, much lower than estimated by the above methods due to ethical concerns and resource constraints. This may be one reason why such approaches have not yet been implemented widely in preclinical research.“

– Please comment on how much evidence (how many replication studies/ different approaches exploring the same question) is needed? And who should collate this body of evidence – researchers, clinicians, funders, a committee?

This is indeed an interesting and pressing question. Researchers from the emerging field of meta-analysis in preclinical research will be well suited to collate such evidence for example in systematic reviews. Such analyses could in turn form the basis for investigator brochures that are required for early clinical trials. We outline this as one scenario and admit that there may be different routes how to arrive at the conclusion that there is enough data to start a clinical trial.

“Systematic replication efforts that are decision-enabling are rare in academic preclinical research and have only recently begun to be conducted (Kimmelman et al., 2014). For the decision to finally engage in a clinical trial a systematic review of all experiments is needed. In such a review, evidence should be judged on the validity and reliability criteria discussed here. Ideally, this will form the basis for informative investigator brochures that are currently lacking such decision enabling information (Wieschowski et al., 2018).”